# A Predictive Model for Dysphagia after Ventilator Liberation in Severe Pneumonia Patients Receiving Tracheostomy: A Single-Center, Observational Study

**DOI:** 10.3390/jcm11247391

**Published:** 2022-12-13

**Authors:** Wanho Yoo, Myung Hun Jang, Sang Hun Kim, Jin A. Yoon, Hyojin Jang, Soohan Kim, Kwangha Lee

**Affiliations:** 1Division of Pulmonary, Allergy and Critical Care Medicine, Department of Internal Medicine, Pusan National University Hospital, Busan 49241, Republic of Korea; 2Department of Rehabilitation Medicine, Pusan National University School of Medicine, Busan 49241, Republic of Korea; 3Department of Internal Medicine, Pusan National University School of Medicine, Busan 49241, Republic of Korea

**Keywords:** swallowing, pneumonia, dysphagia, deglutition disorder, tracheostomy, intensive care units

## Abstract

The main purpose of this study was to develop a model predictive of dysphagia in hospital survivors with severe pneumonia who underwent tracheostomy during their hospital stay. The present study included 175 patients (72% male; mean age, 71.3 years) over 5 years. None of these patients had a history of deglutition disorder before hospital admission. Binary logistic regression analysis was performed to identify factors predicting dysphagia at hospital discharge. Dysphagia scores were calculated from β-coefficients and by assigning points to variables. Of the enrolled patients, 105 (60%) had dysphagia at hospital discharge. Factors prognostic of dysphagia at hospital discharge included being underweight (body mass index < 18.5 kg/m^2^), non-participation in a dysphagia therapy program, mechanical ventilation ≥ 15 days, age ≥ 74 years, and chronic neurologic diseases. Underweight and non-participation in a dysphagia therapy program were assigned +2 points and the other factors were assigned +1 point. Dysphagia scores showed acceptable discrimination (area under the receiver operating characteristic curve for dysphagia 0.819, 95% confidence interval: 0.754–0.873, *p* < 0.001) and calibration (Hosmer–Lemeshow chi-square = 9.585, with df 7 and *p* = 0.213). The developed dysphagia score was predictive of deglutition disorder at hospital discharge in tracheostomized patients with severe pneumonia.

## 1. Introduction

Advances in life-saving medical resources in critically ill patients have resulted in prolonged periods of mechanical ventilation (MV). This is increasing the requirement for tracheostomy. The advantages of tracheostomy include improvements in lung mechanics facilitating oral care, attenuated painful irritation on the larynx/trachea area, reduced requirement for analgesics and sedatives, improved ability to communicate, and reduced time in the intensive care unit (ICU) [1,2,3,4]. Tracheostomized patients, however, frequently experience dysphagia, or deglutition disorder [5,6], which may persist until hospital discharge.

Dysphagia is highly prevalent in post-extubated patients after oral endotracheal intubation [7,8], even if there was no preexisting dysphagia before hospital admission [9]; this is associated with anatomic and functional change during invasive MV [10,11]. Moreover, its incidence is further increasing in tracheostomized patients [12,13]. These patients run the risk of experiencing adverse health outcomes, including increased hospital stay, hospital-acquired pneumonia, and higher long-term mortality rates [8,14,15]. Even in the absence of the lack of a deglutition problem before hospital admission, dysphagia while hospitalized has a detrimental effect on functional outcomes, especially in chronic critically ill patients with severe pneumonia who require a prolonged period of invasive ventilator care and subsequently tracheostomy.

Early identification of dysphagia is essential for early treatment planning, with many studies assessing risk factors for dysphagia following tracheostomy [8,16]. We hypothesized that a model based on these clinical variables would be useful for predicting dysphagia at hospital discharge in tracheostomized patients. The present study evaluated factors predictive of dysphagia in hospital survivors who had been diagnosed with severe pneumonia, received invasive ventilator care, and underwent a tracheostomy during their stay in the ICU. These factors were used to develop a predictive model (dysphagia score), and the ability of this model to predict long-term mortality after hospital discharge was evaluated.

## 2. Materials and Methods

### 2.1. Study Design and Patient Selection

This study, both retrospective and observational, was undertaken in an adult respiratory ICU. This ICU was a 12-bed unit located in the regional center for respiratory diseases, which was established in December 2015. It is part of a tertiary care, university-affiliated 1200-bed hospital. In accordance with therapeutic recommendations, all patients were treated on the basis of a “lung-protective ventilator strategy” [17]. There were no documented criteria for tracheostomy in our country. The decision regarding tracheostomy was mostly handled by critical care physicians when patients were expected to be receiving long-term ventilator care. Additionally, thoracic surgeons performed tracheostomies in the ICU in accordance with standard surgical principles. All hospitalized patients were able to consult with a physiatrist and had access to full time respiratory therapy and physical rehabilitation. For the patients who were confirmed as having swallowing problem after instrumental swallowing assessment, a conventional dysphagia therapy program including suprahyoid muscle strengthening exercise and postural changes/compensatory maneuvers was applied along with neuromuscular electrical stimulation (VitalStim®,Chattanooga Group, Hixson, TN, USA) to the submental muscles by the physiatrists and occupational therapist. In addition, appropriate diet modification was prescribed to prevent food or liquid aspiration into the airway. Dysphagia rehabilitation is a serious concern; however, this is to be carefully deliberated over by the physiatrists and occupational therapists in charge of the case. Based on the patients’ conditions, comorbidities and so forth, they will be able to make an informed decision about how effective this program may or may not be.

Data from adult patients aged ≥18 years were retrospectively evaluated. The inclusion period was from 1 December 2015 to 30 November 2020. To evaluate the 1-year mortality rates after ICU admission, the patient survival status was evaluated until 10 March 2022. Patients were included if (1) they had received ventilator care for more than 24 h; (2) pneumonia, including both community- and hospital-acquired pneumonia was the main diagnosis for MV at ICU admission; (3) they underwent tracheostomy during hospital stay; (4) they did not have a documented deglutition dysfunction before hospital admission; and (5) they survived the hospital stay. The purpose of this study is to develop a predictive model for dysphagia and evaluate the ability of the model to predict long-term mortality. Unfortunately, in order to design this model effectively and attempt to predict long-term mortality rates, we needed to assess surviving patients. Patients with intact oropharyngeal deglutition before ICU admission were included, even if they had underlying chronic neurologic diseases, such as cerebrovascular accidents, intracerebral hemorrhage, and Alzheimer dementia. Patients were excluded if they had undergone tracheostomy or had an irreversible deglutition dysfunction (e.g., on nasogastric tube feeding) before ICU admission. In addition, two patients who had cerebrovascular accidents during their hospital stay were excluded.

In the present study, the swallowing status was analyzed by the result of instrumental swallowing evaluation scores, including videofluoroscopic swallowing study (VFSS) or fiberoptic endoscopic evaluation of swallowing (FEES) performed by the physiatrists. Grading of this severity was then recorded using the penetration aspiration score, whereby a score of above 2 was diagnosed as having swallowing difficulties [18]. These results were gathered during a definite material penetration/aspiration evaluation. The primary study outcome was a deglutition status at hospital discharge, and the secondary outcome was the mortality rate 1-year after ICU admission.

### 2.2. Data Collection

From each subject, clinical variables were retrospectively collected from the electronic medical records (EMRs). We collected the following clinical factors: demographic and clinical data, including age, gender, body mass index (BMI), length of stay (LOS) in the ICU and hospital, and duration of MV. By using the acute physiology and chronic health evaluation (APACHE) II score, the illness severity was measured. Accompanying organ failure was assessed using the sequential organ failure assessment (SOFA) score [19,20], which were both computed using clinical and laboratory data collected within the first 24 h of ICU admission. During the first 24 h of ventilator care, requirements for neuromuscular blocking agents and vasopressors were also recorded. The underlying comorbidities before admission were obtained from EMRs of all enrolled patients. Additionally, we calculated Charlson’s weighted index using their investigated comorbidities [21]. The mortality 1 year after ICU admission was collected from the National Health Insurance Service Database.

Other factors evaluated included the time from the initiation of ventilator care to tracheostomy; patient participation in dysphagia therapy programs directed by physiatrists and physiotherapist or speech language therapists; and the recovery of deglutition functions at hospital discharge. Additionally, we gathered total and out-of-pocket medical expenditures for all medical resources (including all medicines) during the hospital (ICU and general ward) stay from patient EMRs. Expenditures are reported in USD based on the exchange rate on 10 March 2021 of USD 1 equal to KRW 1140.10.

### 2.3. Statistical Analysis

The data collected were continuous data, with normal distribution. This was presented as mean ± standard deviation (SD) and then compared by Student *t*-tests. The variables with non-normal distribution were presented as the median (range) and then compared by Wilcoxon rank-sum tests. Categorical variables were reported as numbers (percentages) and compared by χ2 or Fisher’s exact test, as appropriate. Factors predictive of dysphagia status at hospital discharge on univariate logistic regression analysis were entered into a multivariate logistic regression model, and factors independently predictive of dysphagia status at hospital discharge were determined by backward stepwise logistic regression. The β-coefficients derived from multiple regression analysis were simplified as natural numbers >0; these factors were calculated as the sum of simplified β-coefficients, as described [22,23]. Hence, the predictive model for dysphagia (dysphagia score) was based on the sum of β coefficients. Model discrimination was determined by measuring the areas under the curve (AUCs), and model calibration was analyzed by the Hosmer–Lemeshow test. Kaplan–Meier estimates of 1-year mortality were stratified according to this predictive model and curves were compared using log-rank tests. An optimal cutoff value for the dysphagia score was determined based on the maximum Youden’s index [24]. The sensitivity (SS), specificity (SP), positive likelihood ratio (PLR), negative likelihood ratio (NLR), positive predictive value (PPV), and negative predictive value (NPV) of this cutoff for predicting dysphagia were determined. The AUC of the model was compared with the AUCs of APACHE II and SOFA scores using the DeLong test [25]. All tests were two-tailed, and statistical significance was obtained by a *p*-values of 0.05. All statistical analyses were applied using IBM SPSS ver. 24.0 (IBM Corp., Armonk, NY, USA) and MedCalc Ver.20.109 (MedCalc Software, Ostend, Belgium) statistical software.

## 3. Results

### 3.1. Characteristics of Total Patients

During of the study period, 1746 patients were admitted to the respiratory ICU. After applying inclusion and exclusion criteria, 175 patients were determined to be qualified to participate in the study (Figure 1). The baseline characteristics and clinical outcomes are shown in Table 1. Chronic lung diseases, such as chronic obstructive pulmonary disease, interstitial lung disease, and destroyed lung due to various causes, were the most prevalent underlying comorbidities. The mean period from endotracheal intubation to tracheostomy was 6.1 ± 3.3 days. Of the 175 patients, 105 (60%) had dysphagia at hospital discharge and 77 (44%) died within 1 year.

A comparison of the 105 patients with dysphagia and the 70 patients without dysphagia showed that the former were older and had a significantly lower BMI, a longer duration of MV, a higher rate of chronic neurologic diseases as comorbidities, and a lower rate of participation in dysphagia therapy programs (Table 2).

### 3.2. Predictive Factors Associated with Dysphagia

Table 3 shows the results of multivariate analysis of factors independently predictive of dysphagia at hospital discharge. Five factors were found to be significant: BMI < 18.5 kg/m^2^, non-participation in dysphagia therapy programs, MV LOS ≥ 15 days, age ≥ 74 years, and chronic neurologic diseases as comorbidities. The cut-offs of two of these factors, MV LOS (AUC: 0.610; 95% CI: 0.534–0.683; *p* = 0.010, SS: 58.1%, SP: 64.3%) and age (AUC: 0.603; 95% CI: 0.526–0.676; *p* = 0.019, SS: 55.2%, SP: 67.1%), were based on Youden’s index.

### 3.3. A predictive Model for Dysphagia at Hospital Discharge

A dysphagia score was calculated on the basis of five variables (BMI < 18.5 kg/m^2^ and non-participation in dysphagia therapy programs assigned two points each, and MV LOS ≥ 15 days, age ≥ 74 years, and chronic neurologic diseases as comorbidities assigned one point each), all observed in multivariate analyses in accordance with the β coefficients. This model had excellent discrimination, with an AUC for predicting dysphagia of 0.819 (95% CI: 0.754–0.873; Figure 2) and excellent calibration (Hosmer–Lemeshow chi-square = 9.585, with df = 7 and *p* = 0.213). The number of patients for the dysphagia score and corresponding dysphagia rate at hospital discharge (%) was shown in Figure 2. The cut-off value for the dysphagia score based on the maximum Youden index was ≥3 (SS, 56%: SP, 93%; PLR, 7.8; NLR, 0.47; PPV, 92.2%; NPV, 58.6%). Of the 64 patients with dysphagia score ≥3, 59 (92.2%) had dysphagia at hospital discharge. In addition, compared to the AUC for the APACHE II and SOFA scores, the dysphagia score’s AUC was much greater (Figure 3).

### 3.4. Predicting Long-Term Mortality Using the Dysphagia Score

Although all enrolled patients survived hospitalization, their 90-, 180-, and 365-day mortality rates after ICU admission were 20.6%, 32.0%, and 44.0%, respectively. The AUCs of dysphagia scores for predicting 90-, 180-, and 365-day mortality were 0.683 (95% CI: 0.597–0.769, *p* = 0.001), 0.661 (95% CI: 0.579–0.744, *p* = 0.001), and 0.637 (95% CI: 0.555–0.719, *p* = 0.001), respectively. The 90-, 180-, and 365-day mortality rates in the 64 patients with dysphagia scores ≥3 were 34.4%, 46.9%, and 57.8%, respectively. Figure 4 shows the Kaplan–Meier survival curves based on this score for 365-day cumulative mortality rate (Log-rank 53.163, *p* < 0.001).

## 4. Discussion

The present study evaluated factors predictive of oropharyngeal dysphagia in hospital survivors with severe pneumonia who underwent surgical tracheostomy during their hospital stay. This analysis identified five factors independently predictive of dysphagia at hospital discharge: older age, being underweight, longer duration of invasive mechanical ventilation, chronic neurologic diseases as comorbidities, and non-participation in dysphagia therapy programs. A predictive model was developed using these indicators, with the resulting dysphagia score showing good discrimination and calibration for predicting dysphagia. Moreover, this score was better able to predict dysphagia than APACHE II and SOFA scores at ICU admission. These scores were based on objective criteria and were easy to calculate from parameters recorded in patients’ EMRs. Systematic screening at the bedside of patients with higher scores may lead to early introduction of rehabilitative measures, as well as providing information to patients’ families and surrogates about the risks of oropharyngeal dysphagia after tracheostomy.

Because the factors predictive of dysphagia have also been reported to be prognostic factors associated with long-term prognosis in critically ill patients [22,26,27], the calculated dysphagia scores may also be predictive of long-term patient mortality. Although this model would be worthy of reference when there is discussion between physicians and patients’ surrogates about future care plan, the AUC of this model was lower for long-term mortality than for dysphagia. The utility of the evolution of this model over time would be interesting and should be confirmed in future study for predicting long-term mortality.

Although few studies have evaluated the treatment of dysphagia [7], several therapeutic interventions have been utilized to improve the deglutition function in tracheostomized patients [8,28,29,30,31]. The results of the present study also showed that participation in dysphagia therapy programs was significantly associated with recovery from dysphagia. Patients who undergo tracheostomy in the ICU should be routinely screened for dysphagia. Moreover, additional studies are required to determine optimal dysphagia rehabilitation methods in these tracheostomized patients [7,8,16,32].

Consistent with previous findings [2,8,33,34], the present study found that the rate of dysphagia was remarkably higher in tracheostomized hospital survivors, even in the absence of a deglutition disorder before ICU admission. In addition, 69.7% of all included patients underwent tracheostomy within 7 days after invasive MV. Our findings raise important considerations. Relative to their sizes, many tertiary care hospitals in Asian countries (including South Korea) have a small number of ICU beds [35], with this shortage limiting the ability to treat acutely ill patients. In addition, when there were tracheostomized patients with lower inspiratory pressure and oxygen requirements, they would be placed in the general ward on a home ventilator system, which would be helpful for controlling the incidence of accompanying delirium. As a consequence, the decision-making for tracheostomy would be earlier in patients with a higher likelihood of survival during their hospital stay. Earlier tracheostomy may benefit patients with a higher chance of survival during their ICU stay, but critical care physicians should be made aware of the likelihood of dysphagia following tracheostomy in patients receiving long-term ventilator care.

The present study had a number of limitations. First, the factors found to be predictive of dysphagia in the present study had previously been identified as risk factors for dysphagia in post-extubation and tracheostomized patients [7,8,16]. Although we hypothesized that other clinical factors may be risk factors for dysphagia, we identified no additional risk factors due to the retrospective design of our analysis. Second, the cut-off value of dysphagia score had low SS and NPV, which may be due to the relatively small number of enrolled patients. Additional long-term studies in larger numbers of patients are needed to validate this model for predicting both dysphagia and long-term mortality. Third, this study included patients hospitalized in a single ICU, suggesting that the data may not be generalize to other hospitals.

## 5. Conclusions

Based on existing studies, this is the first known study to generate a model that can predict deglutition dysfunction at hospital discharge in tracheostomized patients with severe pneumonia. Use of this model could result in the early identification of high-risk patients and earlier introduction of rehabilitative measures. Large-scale multicenter studies are needed to develop a useful model predicting dysphagia in tracheostomized patients.

## Figures and Tables

**Figure 1 jcm-11-07391-f001:**
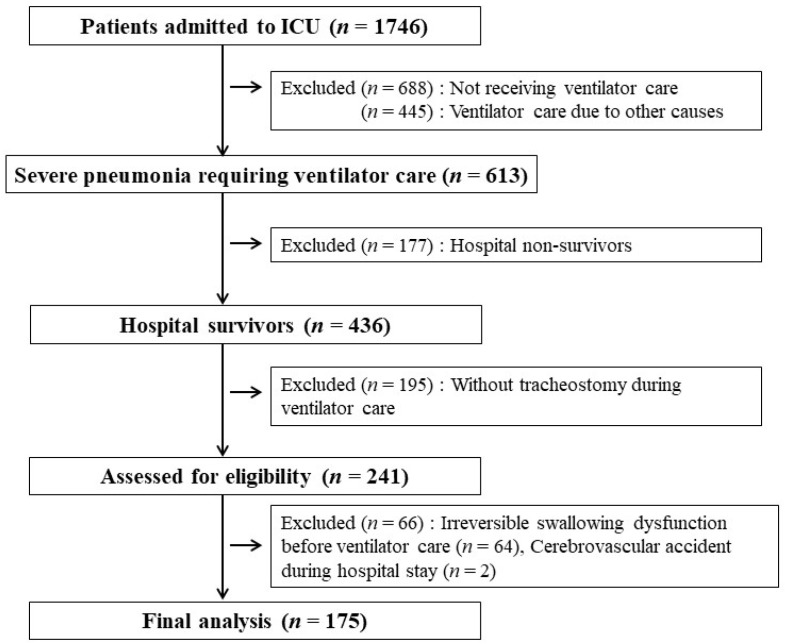
Flowchart of patient selection.

**Figure 2 jcm-11-07391-f002:**
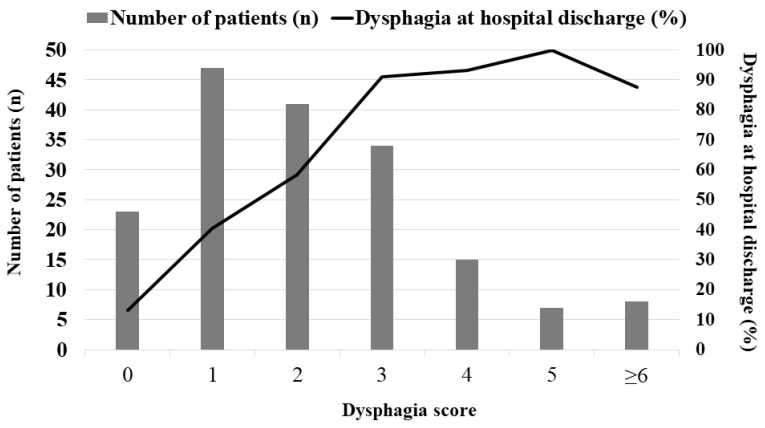
The number of patients for the dysphagia score and corresponding dysphagia rate at hospital discharge.

**Figure 3 jcm-11-07391-f003:**
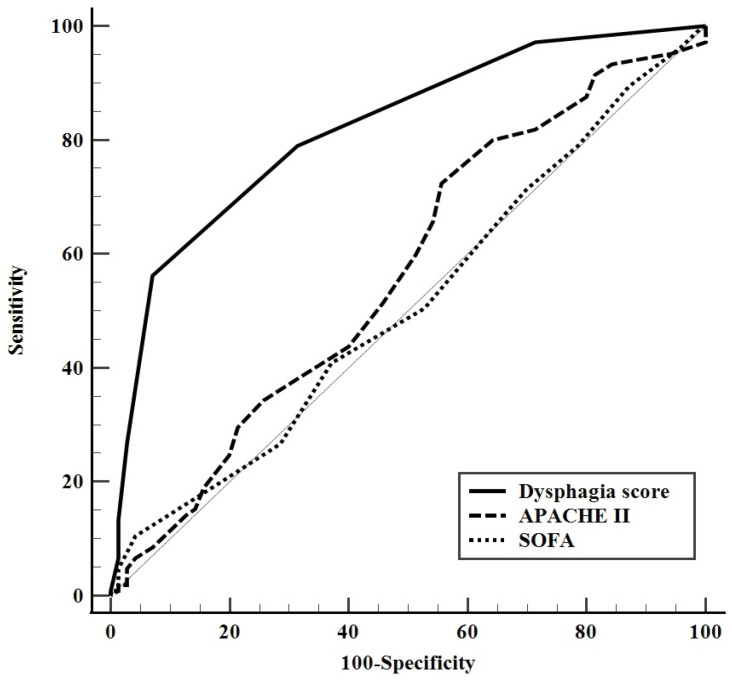
Comparison of receiver operating characteristic (ROC) curves for dysphagia score, Acute Physiology and Chronic Health Evaluation (APACHE) II score, and Sequential Organ Failure Assessment (SOFA) score for predicting dysphagia at hospital discharge. For all patients, the areas under the ROC curves (AUCs) for dysphagia score, APACHE II score, and SOFA score were 0.819 (95% confidence interval [CI]: 0.754–0.873; *p* < 0.001), 0.568 (95% CI: 0.491–0.643; *p* = 129), and 0.511 (95% CI: 0.435–0.587; *p* = 0.891), respectively. The AUC for dysphagia score was significantly higher than the AUCs for the other two scores (*p* < 0.001 each).

**Figure 4 jcm-11-07391-f004:**
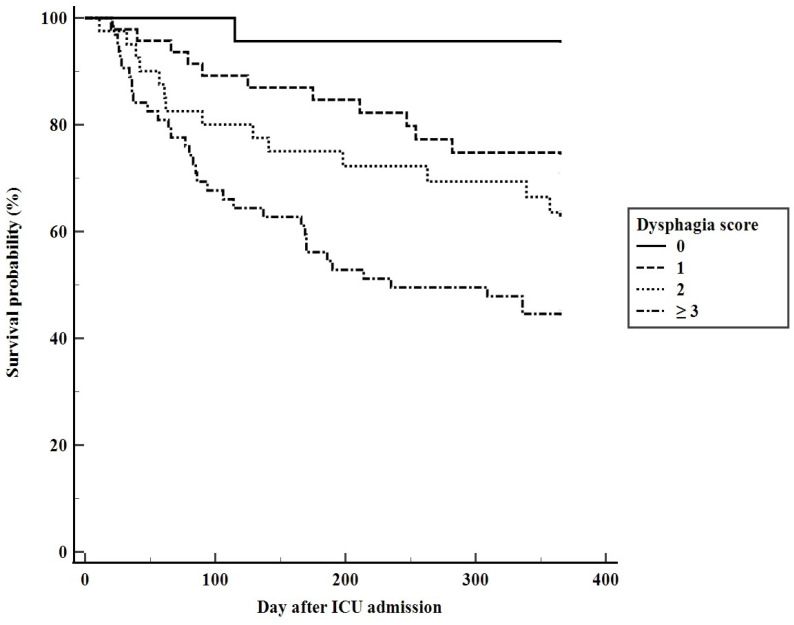
Kaplan–Meier analyses of patient survival curves 1 year after ICU admission as a function of dysphagia score.

**Table 1 jcm-11-07391-t001:** Clinical characteristics of total patients.

	Total (*n* = 175)
Male	126 (72.0%)
Age (years)	71.3 ± 10.6
BMI (kg/m^2^)	21.6 ± 3.7
ICU LOS (days)	17.4 ± 9.8
MV LOS (days)	15.7 ± 9.4
Hospital LOS (days)	35 (15–54)
APACHE II score *	20.8 ± 6.3
SOFA score *	7.5 ± 3.1
Charlson’s comorbidity index	2.3 ± 1.6
Total medical expenditures (USD) ^†^	27,737 (4832–100,065)
Out-of-pocket medical expenditures (USD) ^†^	6634 (518–65,591)
Period from endotracheal intubation to tracheostomy (days)	6.2 ± 3.4
Dysphagia at hospital discharge	105 (60.0%)
One-year cumulative mortality after ICU admission	77 (44.0%)

We can use continuous data with normal distribution and present it as mean ± standard deviation (SD); conversly, continuous data combined with non-normally distribution are presented as median (range), and categorical variables are reported as numbers (%). * Both scores were both computed using clinical and laboratory data within the first 24 h of ICU admission. ^†^ Total and out-of-pocket medical expenditures for all medical resources used (including all medicines) during stay in the ICU and general ward. Abbreviations: BMI, body mass index; ICU, intensive care unit; MV, mechanical ventilation; LOS, length of stay; APACHE II, Acute Physiology and Chronic Health Evaluation II; SOFA, Sequential Organ Failure Assessment; USD, U.S. dollars.

**Table 2 jcm-11-07391-t002:** Clinical characteristics of patients with and without dysphagia at hospital discharge.

	Dysphagia at Hospital Discharge		
	Yes (*n* = 105)	No (*n* = 70)	*p*-Value
Male	73 (69.5)	53 (75.7)	0.396
Age, yr	72.7 ± 10.1	69.1 ± 11.0	0.003
BMI, kg/m^2^	20.8 ± 3.6	23.0 ± 3.6	<0.001
APACHE II score *	21.3 ± 6.2	20.0 ± 6.4	0.167
SOFA score *	7.6 ± 3.1	7.4 ± 3.0	0.726
MV LOS	17.2 ± 10.1	13.6 ± 7.8	0.013
Comorbidities			
Chronic lung diseases ^†^	41 (39.0)	23 (32.9)	0.427
Cardiovascular diseases	36 (34.3)	20 (28.6)	0.509
Chronic neurological diseases ^‡^	41 (39.0)	15 (21.4)	0.020
Diabetes	26 (24.8)	23 (32.9)	0.303
Solid malignant tumors	11 (10.5)	14 (20.0)	0.121
Chronic kidney diseases	10 (9.7)	9 (12.9)	0.621
Requirement for neuromuscular blocking agents during the first 24 h after mechanical ventilation	69 (65.7)	53 (75.7)	0.181
Requirement for vasopressors during the first 24 h after mechanical ventilation	64 (61.0)	40 (57.1)	0.640
Period from endotracheal intubation to tracheostomy (days)	6.4 ± 3.5	5.8 ± 3.1	0.225
Dysphagia therapy programs during hospital stay ^§^	68 (64.8)	64 (91.4)	<0.001
One-year cumulative mortality	60 (57.1)	17 (24.3)	<0.001

We can use continuous data with normal distribution and present it as mean ± standard deviation (SD); conversly, continuous data combined with non-normal distribution are presented as median (range); and categorical variables are reported as numbers (%). * Both scores were both computed using clinical and laboratory data within the first 24 h of ICU admission. ^†^ Including chronic obstructive pulmonary disease, interstitial lung disease, and destroyed lung due to different causes. ^‡^ Including cerebrovascular accidents, intracerebral hemorrhage, subdural hemorrhage, subarachnoid hemorrhage, and Alzheimer dementia. ^§^ Including suprahyoid muscle strengthening exercise, postural changes/compensatory maneuvers, and appropriate diet modification by physiatrists and occupational therapists designed to improve swallowing function. Abbreviations: BMI, body mass index; MV, mechanical ventilation; LOS, length of stay; APACHE II, Acute Physiology and Chronic Health Evaluation II; SOFA, Sequential Organ Failure Assessment; USD, U.S. dollars; ICU, intensive care unit.

**Table 3 jcm-11-07391-t003:** Multivariate analyses of factors associated with dysphagia at hospital discharge.

Variables	Adjusted OR * (95% CI)	*p*-Value	β Value
BMI < 18.5 kg/m^2^	11.219 (2.917–43.142)	<0.001	2.418
No dysphagia therapy programs	5.546 (2.000–15.380)	0.001	1.713
MV LOS ≥ 15 days ^†^	3.776 (1.766–8.075)	0.001	1.329
Age ≥ 74 years ^†^	3.445 (1.608–7.382)	0.001	1.237
Chronic neurologic diseases as comorbidities ^‡^	2.485 (1.096–5.636)	0.029	0.910

* All factors were indentified by the multivariate analysis using stepwise backward selection procedures (Hosmer–Lemeshow chi-square = 9.585, df = 7, *p* = 0.213). ^†^ All optimal cut-off levels were based on Youden’s index (AUC: 0.610, 95% CI: 0.534–0.683, *p* = 0.010, sensitivity: 58%, specificity: 64%, for MV LOS; AUC: 0.603; 95% CI: 0.526–0.676, *p* = 0.019, sensitivity: 55%, specificity: 67%, for age). ^‡^ Including cerebrovascular accidents, intracerebral hemorrhage, subdural hemorrhage, subarachnoid hemorrhage, and Alzheimer dementia. Abbreviations: BMI, body mass index; MV, mechanical ventilation; LOS, length of stay; AUC, area under the receiver operation characteristic curves; CI, confidence interval.

## Data Availability

The data presented in this study are available upon request from the corresponding author. The data are not publicly available due to privacy regulations.

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
