# Peer review of "A Predictive Model for Dysphagia after Ventilator Liberation in Severe Pneumonia Patients Receiving Tracheostomy: A Single-Center, Observational Study"

_jcm, 2022, doi:10.3390/jcm11247391_

Round 1

Reviewer 1 Report

This paper presents a study about a very important issue, dysphagia post-ICU treatment. The introduction is comprehensive, the methods are well presented and the results are well presented and meaningful.

A major question arising when one is reading this paper is about the protocols applied regarding the decision to proceed to tracheo in ICU-treated patients (with the diagnoses concerning the present study). Since the authors focus their research on ICU-treated tracheotomized patients without pre-existing dysphagia and the time of tracheotomy that they report is rather early some obvious questions occur: what were the criteria for thacheostomy in their patients; did they compare tracheotomized and non-tracheotomized patients?

Recent literature has addressed the factors contributing to ICU-related dysphagia and they     should be discussed in general terms since the authors set about defining a predictive model. Relevant papers include:

Scheel R, et al. Endoscopic Assessment of Swallowing After Prolonged Intubation in the ICU Setting. Ann Otol Rhinol Laryngol, 2016.

Printza A, et al. Dysphagia Severity and Management in Patients with COVID-19. Curr Health Sci J. 2021

Brodsky M, et al. Coordination of pharyngeal and laryngeal swallowing events during single liquid swallows after oral endotracheal intubation for patients with acute respiratory distress syndrome. Dysphagia, 2018.

Langmore SE,et al. Abnormalities of Aspiration and Swallowing Function in Survivors of Acute Respiratory Failure. Dysphagia, 2020

More details regarding dysphagia rehabilitation are also needed ( since this is a contributor to the predictive model) especially about the reasons of not having anything (not even dietary changes): was this due to the severity of the patients condition, comorbidities, non-availability of swallowing treatment?

Minor comments

Keywords: add swallowing; pneumonia; dysphagia

Introduction, line 40: Despite rephrase to “Even in the absence”

Results

Table 3. Footnotes: Lines 182-185 can be omitted. The information is in the text.

Results-Discussion. The expediture results are not commented. Either comment them or do not present them if they are irrelevant.

Author Response

Response to reviewer’s comment (Reviewer #1)

This paper presents a study about a very important issue, dysphagia post-ICU treatment. The introduction is comprehensive, the methods are well presented and the results are well presented and meaningful.

Reply] Thank you for taking the time to review our paper. We revised our paper according the reviewer’s comments.

A major question arising when one is reading this paper is about the protocols applied regarding the decision to proceed to tracheo in ICU-treated patients (with the diagnoses concerning the present study). Since the authors focus their research on ICU-treated tracheotomized patients without pre-existing dysphagia and the time of tracheotomy that they report is rather early some obvious questions occur: what were the criteria for thacheostomy in their patients; did they compare tracheotomized and non-tracheotomized patients?

Reply] Thank you for your comments.

With respect to “the protocols applied regarding the decision to proceed to tracheostomy in ICU treated patients”, this was solely at the discretion of the critical care physicians handling those patients. Thus, the criteria for tracheostomies in the ICU patients were not documented as these were made at the discretion of the critical care physicians. Furthermore, no comparison was made between those tracheostomized and not as it would be steer from the purpose of our study. We added some sentences in method section.

à Page 2, There were no documented criteria for tracheostomy in our country. Decision regarding tracheostomy was mostly handled by critical care physicians when patients were expected to be receiving long-term ventilator care

Recent literature has addressed the factors contributing to ICU-related dysphagia and they     should be discussed in general terms since the authors set about defining a predictive model. Relevant papers include:

Scheel R, et al. Endoscopic Assessment of Swallowing After Prolonged Intubation in the ICU Setting. Ann Otol Rhinol Laryngol, 2016.

Printza A, et al. Dysphagia Severity and Management in Patients with COVID-19. Curr Health Sci J. 2021

Brodsky M, et al. Coordination of pharyngeal and laryngeal swallowing events during single liquid swallows after oral endotracheal intubation for patients with acute respiratory distress syndrome. Dysphagia, 2018.

Langmore SE,et al. Abnormalities of Aspiration and Swallowing Function in Survivors of Acute Respiratory Failure. Dysphagia, 2020

Reply] Thank you for your comments. The recommended papers were reviewed. In comparison to our study, these studies mainly enrolled post-extubated patients after oral endotracheal intubation. Therefore, it was difficult to add some content in the discussion section. Instead, we added some sentences in the introduction section rather than in in the discussion section.  

à Page 1, Dysphagia is highly prevalent in post-extubated patients after oral endotracheal in-tubation [7,8], even if there was no preexisting dysphagia before hospital admission [9]; this is associated with anatomic and functional change during invasive MV [10,11]. Moreover, its incidence is further increasing in tracheostomized patients [12,13].   

More details regarding dysphagia rehabilitation are also needed ( since this is a contributor to the predictive model) especially about the reasons of not having anything (not even dietary changes): was this due to the severity of the patients condition, comorbidities, non-availability of swallowing treatment?

Reply] Thank you for your comments. We revised some sentences with detailed description about dysphagia rehabilitation in the method section.

à Page 2. For the patients who were confirmed as having swallowing problem after instrumental swallowing assessment, conventional dysphagia therapy program including suprahyoid muscle strengthening exercise, postural changes/compensatory maneuvers were applied along with neuromuscular electrical stimulation (VitalStim TM) to the submental muscles by the physiatrists and occupational therapist. In addition, appropriate diet modification was prescribed to prevent food or liquid aspiration into the airway. Dysphagia rehabilitation is a serious concern, however, this is to be carefully deliberated over by the physiatrists and occupational therapists in charge of the case. Based on the patients’ conditions, comorbidities and so forth, they will be able to make an informed decision about how effective this program may or may not be.

Minor comments

Keywords: add swallowing; pneumonia; dysphagia

Reply] We added these keywords in abstract.

Introduction, line 40: Despite rephrase to “Even in the absence”

Reply] We revised this sentence according to comments.

à Page 1-2. Even in the absence ~

Results

Table 3. Footnotes: Lines 182-185 can be omitted. The information is in the text.

Reply] We omitted indicated lines in footnote of Table 3.

Results-Discussion. The expediture results are not commented. Either comment them or do not present them if they are irrelevant.

Reply] We revised contents regarding the expenditure in method, and results section. We explained the expenditure in only Table 1.

Reviewer 2 Report

This study is interesting. However, the manuscript requires some corrections.

(1) Distribution histogram of dysphagia score should be shown.

(2) Diagnosis of dysphasia is not clear although the authors described something in the text. Severity of dysphasia should be considered, and grading based on the severity is necessary for diagnosis.

Author Response

Response to reviewer’s comment (Reviewer #2)

This study is interesting. However, the manuscript requires some corrections.

(1) Distribution histogram of dysphagia score should be shown.

Reply] Thank you for taking the time to review our paper. We added a figure regarding histogram of dysphagia score and a sentence according to reviewer’s comment.

à Page 7, The number of patients for dysphagia score and corresponding dysphagia rate at hospital discharge (%) was shown Figure 2.

(2) Diagnosis of dysphasia is not clear although the authors described something in the text. Severity of dysphasia should be considered, and grading based on the severity is necessary for diagnosis.

Reply] We revised some sentences regarding diagnosis of dysphagia in method section according to reviewer’s comment.

à Page 3 It should be noted that diagnoses of dysphagia were omitted from this study as they were left entirely to the critical care physicians discretion. The severity of dysphagia was recorded using the analyses of instrumental swallowing evaluation scores, including videofluoroscopic swallowing study (VFSS) or fibroptic endoscopic evaluation of swallowing (FEES) performed by the physiatrists. Grading of this severity was then recorded using the penetration aspiration score, whereby, a score of above 2 was diagnosed as having swallowing difficulties. These results were gathered during a definite material penetration/aspiration evaluation.

Reviewer 3 Report

The topic of this article is interesting. However, i think there has a big flaw in the study design, which significantly limits the clinical application. As the aim of this study is to bulid a model for predicting dysphagia in tracheostomized patients, they excluded patients who died during hospitalization. I don't know how to use this model in clincal practice and guide early interventions. It's very hard to distinguish alive or died patients on admission or at any time point during hospital stay, expect the event happens. Therefore, the authors must clearly explain this problem.   

Author Response

Response to reviewer’s comment (Reviewer #3)

The topic of this article is interesting. However, i think there has a big flaw in the study design, which significantly limits the clinical application. As the aim of this study is to bulid a model for predicting dysphagia in tracheostomized patients, they excluded patients who died during hospitalization. I don't know how to use this model in clincal practice and guide early interventions. It's very hard to distinguish alive or died patients on admission or at any time point during hospital stay, expect the event happens. Therefore, the authors must clearly explain this problem.

Reply] Thank you for taking the time to review our paper.

The purpose of this study is to develop a predictive model for dysphagia and evaluate the ability of the model to predict long-term mortality. Unfortunately, in order to design this model effectively and attempt to predict long-term mortality rates, we needed to assess surviving patients. There are a variety of indicators used for predicting dysphagia, which may also be applicable for prognostic indicators in the critical care field. If we considered non-survivors, there would be too many variables, thereby making the study too broad and difficult to use as a tool. We add some sentence in the methods section.

à Page 2. The purpose of this study is to develop a predictive model for dysphagia and evaluate the ability of the model to predict long-term mortality. Unfortunately, in order to design this model effectively and attempt to predict long-term mortality rates, we needed to assess surviving patients

Round 2

Reviewer 2 Report

Study setting that dysphagia diagnosis was made by decision of each physician only is not good. It requires uniform criteria for scientific study. There develops arbitrary interference. Was swallowing evaluation score used for diagnosis?

Author Response

Response to the comment (reviewer #2)

Study setting that dysphagia diagnosis was made by decision of each physician only is not good. It requires uniform criteria for scientific study. There develops arbitrary interference. Was swallowing evaluation score used for diagnosis?

Reply] Thank you for your comment. Dysphagia diagnosis was made by critical care physician, however, its diagnosis was totally based on an analysis of VFSS or FEES. We also used penetration aspiration score for diagnosis as having swallowing difficulties as described below. We revised some sentences regarding dysphagia diagnosis in the method section.

Page 3 --> In the present study, swallowing status was analyzed by the result of instrumental swallowing evaluation scores, including videofluoroscopic swallowing study (VFSS) or fibroptic endoscopic evaluation of swallowing (FEES) performed by the physiatrists. Grading of this severity was then recorded using the penetration aspiration score, whereby, a score of above 2 was diagnosed as having swallowing difficulties. These results were gathered during a definite material penetration/aspiration evaluation.

Reviewer 3 Report

No additional comments

Author Response

No response to reviewer’s comment because reviewer has no additional comments